# OpenReview forum: "UniDetox: Universal Detoxification of Large Language Models via Dataset Distillation"
_ICLR.cc/2025/Conference — ICLR 2025 Poster_

### Official Review · Reviewer_zPHB · 2024-10-19

**Soundness:** 3
**Presentation:** 4
**Contribution:** 3
**Rating:** 8
**Confidence:** 3

**Summary:**

The research presents a novel and universally applicable detoxification approach to a wide range of language models without the prior understanding or fine-tuning of the specific hyperparameters and architectures. The work introduces an efficient data distillation method by incorporating contrastive decoding, which helps to identify the toxicity vector within the initial parameter space and generate detoxifying text sets. Later, the language model is fine-tuned on the distilled datasets to mitigate the toxicity in the base language model. The approach was evaluated and analyzed based on various baseline methods and UniDetox to understand the technique's effectiveness. Overall, the method, UniDetox, introduces a novel data distillation algorithm with contrastive decoding to significantly mitigate the level of toxicity in the base language models.

**Strengths:**

The work provides a novel approach and insights into the following points
1. Novel dataset distillation method using contrastive decoding
2. Complete evaluation across a range of commonly used detoxification methods from the references
3. Significant results based on the PT and EMT metrics across different benchmarks

As for the paper, the strengths are
1. Well-structured and written paper
2. Well-styled and formatted paper
3. Thoughtful introduction and rigorous reasoning

**Weaknesses:**

1. The size of the model might mitigate UniDetox's effectiveness since the ratio of parameter space and the volume of detoxifying text might be increased significantly, and no further preventive techniques could be introduced.
2. Compared to other methods, UniDetox still requires model fine-tuning to mitigate the toxicity from the base models, making this approach's computational intensity a further challenge or question to understand.
3. Limited to the text-only language models, as the emerging traction of multi-modal language models, how this approach further mitigates toxicity with a vast data modality remains challenging to this technique.

**Questions:**

The following points suggest further investigation and evaluation of this method's potential weaknesses or limitations.
1. Does this method apply to the scope beyond the text-only model? Can this approach apply to the multimodal language models as well?
2. What are the costs regarding the computational resources, time required, or other considerations?
3. What is the impact of the larger parameters space, like using larger models from 14B to even 405B models, on the small set of detoxifying text?

---

> ### Author Response · Authors · 2024-11-21
> **Response to Reviwer zPHB**
>
> Dear Reviewer zPHB,
>
> Thank you for your thoughtful and detailed feedback on our submission.
> We truly appreciate the time and effort you have dedicated to reviewing our work and providing constructive comments.
> We are also grateful for your recognition of the originality and novelty of our approach.
> Below, we address each of your concerns in detail.
>
> ---
>
> # Impact of Model Size on UniDetox's Effectiveness
>
> > **Comment:** The size of the model might mitigate UniDetox's effectiveness since the ratio of parameter space and the volume of detoxifying text might be increased significantly, and no further preventive techniques could be introduced.
> >
> > **Question:** What is the impact of the larger parameters space, like using larger models from 14B to even 405B models, on the small set of detoxifying text?
>
> We appreciate your suggestion to evaluate UniDetox on even larger models (14B to 405B parameters).
> However, due to computational constraints, extending our experiments to models as large as 405B parameters is currently challenging.
> While UniDetox can directly utilize detoxifying text distilled from a smaller GPT-2 XL, implementing baselines like DExperts and Task Arithmetic for 405B models would require fine-tuning toxic variants of these large models, which is prohibitively expensive in terms of time and storage resources.
>
> ---
>
> # Computational Costs of UniDetox
>
> > **Comment:** Compared to other methods, UniDetox still requires model fine-tuning to mitigate the toxicity from the base models, making this approach's computational intensity a further challenge or question to understand.
> >
> > **Question:** What are the costs regarding the computational resources, time required, or other considerations?
>
> We have included a comparison of the computational resources and time required for each method in Table 7 of Appendix A.4 in our paper.
>
> While UniDetox requires fine-tuning to detoxify models, it does not necessitate preparing a toxic variant for each model to be detoxified.
> In practice, we only fine-tune one toxic GPT-2 XL model to implement UniDetox across different target models.
>
> In contrast, methods like DExperts and Task Arithmetic require preparing toxic versions of each target model (e.g., OPT-6.7B, Falcon-7B, LLaMA2-7B) for detoxification.
> Additionally, DExperts incurs extra computational overhead during inference due to its dual forward pass requirement.
>
> Overall, UniDetox is more computationally efficient compared to these approaches.
>
> ---
>
> # Extending UniDetox to Multimodal Language Models
>
> > **Comment:** Limited to text-only language models, as the emerging traction of multimodal language models, how this approach further mitigates toxicity with a vast data modality remains challenging to this technique.
> >
> > **Question:** Does this method apply to the scope beyond text-only models? Can this approach apply to multimodal language models as well?
>
> Yes, theoretically, UniDetox can be extended to multimodal models.
> As long as there is toxic multimodal data available, UniDetox can be applied similarly to how it is used with text-only models.
>
> For image-to-text models, UniDetox can be implemented in the same contrastive decoding fashion as in our study.
> For text-to-image models, as long as the decoding process can be adapted to a contrastive approach, UniDetox can be applied effectively.
>
> ---
>
> We hope that these revisions effectively address your concerns.
> We believe that our approach offers a significant contribution to the field by providing a universally applicable detoxification method for large language models.
>
> Thank you once again for your insightful feedback.
>
> Sincerely,
> *The Authors*

---

> > ### Comment · Reviewer_zPHB · 2024-11-27
> >
> > Understood, these responses are enough to address the concerns I have

---

> > > ### Author Response · Authors · 2024-11-27
> > >
> > > Dear Reviewer zPHB,
> > >
> > > Thank you for taking the time to review our responses and the revised manuscript.
> > >
> > > We truly appreciate your thoughtful feedback and support for our work.
> > >
> > > Sincerely,
> > >
> > > The Authors

---

### Official Review · Reviewer_7CV4 · 2024-10-31

**Soundness:** 3
**Presentation:** 3
**Contribution:** 3
**Rating:** 6
**Confidence:** 5

**Summary:**

This work proposes an approach to detoxify LLMs via fine-tuning (UniDetox), with the novelty of using dataset-distilled exemplars. The authors propose to distill datasets using contrastive decoding. An analysis based in fine-tuning gradients is provided, showing that the proposed fine-tuning strategy is indeed moving the model weights in the opposite direction of the toxicity vector. UniDetox requires (1) fine-tuning a base model (B) to be toxic, obtaining model (T), (2) obtaining distilled text with contrastive decoding according to (B-T) and (3) fine-tuning a target model with such data. The authors claim that the obtained data and fine-tuning parameters are universal (i.e. can be applied to any LLM).
Experiments show better toxicity mitigation than DExperts (Liu et al. 2021) on the source GPT2 model, as well as on other LLMs that are fine-tuned with distilled data using GPT2. Additionally, perplexity and n-gram metrics are provided as proxies for fluency.

**Strengths:**

**Originality:**

* Using dataset distillation in the setting of LLMs in an actionable way is original, to the best of my knowledge.
* The use of contrastive decoding is also an interesting aspect of this work.

**Clarity:**

* The paper is well written, with clear language. The mathematical notation and formulation is also clear and easy to follow. The figures are also self-explanatory and provide good insights on the work.

**Significance:**

* Toxicity mitigation is an extremely important topic, this work can be of interest to the community. However, see weaknesses and questions for suggestions on how to increase the impact.

**Weaknesses:**

**Quality:**

* The experimental setup can be improved to convince the community about the validity of the proposed method. I suggest the authors to include some zer/few-shot metrics (eg. MMLU) to understand how the model abilities are impacted. I also have concerns about the perplexity reported and the performance on OOD data (see questions).
* Ablations:
  * The proposed method has an important hyper-parameter $\lambda$. I missed an ablation study on $\lambda$, showing the impact of this choice in the final performance.
  * The amount of data produced via distillation is of great importance. Can the authors discuss about the data required for UniDetox to be effective? Additionally, I would be really curious in seeing some of the distilled sentences produced by UniDetox, as well as examples generated by the detoxified model. Such qualitative results can help the reader understand the method.

**Clarity:**

* I kindly ask the authors to provide the steps followed to derive the Taylor approximation in Eq. 5 (from $s(x)$ in Eq. 1.

**Significance:**

* The evaluation proposed (methods compared) and metrics (ppl, n-gram) are too weak given the current abilities of LLMs and the thorough evaluations many works provide nowadays.

**Questions:**

* In Section 3.1, I encourage the authors to include some metric related to zero/few-shot abilities. For example, the overall score in MMLU seems a good candidate.

* Results in Table 1. I have some concerns about the methods used for comparison. Although DExperts was a successful method when it was proposed, several methods have shown superior performance in recent years. For example, [SuauICML24] proposes a method to reduce toxicity without any inference cost and not requiring fine-tuning, showing superior performance compared to DExperts or pre-prompting. Similarly, [PozzobonEMNLP23] propose an efficient method for toxicity mitigation.  I encourage the authors to consider comparing with some newer method than DExperts, this work's imact could strongly benefit from that.
I also suggest adding these recent methods in the Related Work.

> [SuauICML24] Suau, X., Delobelle, P., Metcalf, K., Joulin, A., Apostoloff, N., Zappella, L., & Rodríguez, P. (2024). Whispering experts: Neural interventions for toxicity mitigation in language models. ICML 2024.

> [PozzobonEMNLP23] Pozzobon, Luiza, et al. "Goodtriever: Adaptive toxicity mitigation with retrieval-augmented models." EMNLP 2023 Findings.

* Results in Table 1. Could the authors comment on the fact that PPL goes down from 17.28 to 12.23 when using UniDetox? To me, this behavior is counterintuitive, since the PPL of a LLM rarely goes down when one intervenes on the LLM. In this case, fine-tuning on distilled data is an intervention on the original model. It is surprising that fine-tuning on a small synthetic dataset brings the LLM perplexity to 12.23 (which would be the PPL of a much larger/stronger LLM).
I encourage the authors to provide details about the PPL evaluation, and a conclusive justification of why such PPLs are strongly decreasing.

  * Additinally, in Table 2, UniDetox reduces PPL for OPT, Falcon and increases PPL for LLama2. Conversely, using lr=1e-5, the PPL strongly increases for OPT, Falcon and decreases for LLama2. This hints that the fine-tuning parameters and/or distilled data are not _universal_ but rather well suited for the combinations of parameters and models chosen.

* Results in Table 1. Why is OOD toxicity reduction stronger than ID reduction? I would have expected a fine-tuning based approach like UniDetox to be much more effective on ID. Could the authors comment on this aspect?

* Results in Table 1. For the safety preprompt to be effective, have the authors considered evaluating an instruction-tuned model such as gemma-2-2b-it? I believe safety preprompts are much better designed for instruction tuned models than for decoder only (and arguably older) models like GPT2.

**Comments**:

* This work has some interesting proposals such as using dataset distillation to address model alignment. However, the _universality_ claims are only validated using a small number of models and a non-comprehensive set of metrics. I find very confusing the fact that PPL strongly decreases when the intervention is applied. Moreover, details about the choice of $\lambda$ and the data produced are lacking. Considering all the above, I cannot recommend this paper for acceptance as is, however, I am open to reconsider my score upon rebuttal.

---

> ### Author Response · Authors · 2024-11-21
> **Response to Reviewer 7CV4 1/4**
>
> Dear Reviewer 7CV4,
>
> Thank you for your thoughtful and detailed feedback on our submission. We sincerely appreciate the time and effort you have invested in reviewing our work and providing constructive comments. We are also grateful for your recognition of the originality and novelty of our approach. Below, we address each of your concerns in detail.
>
> ---
> # Derivation of the First-Order Taylor Approximation
>
> > **Question:** I kindly ask the authors to provide the steps followed to derive the Taylor approximation in Eq. 5 (from $s(\mathbf{x})$ in Eq. 1.
>
> Specifically, we expand $\log p_{\theta_\mathrm{toxic}}(x)$ around $\log p_{\theta_\mathrm{base}}(x)$
>
> $$
> \log p_{\theta_\mathrm{toxic}}(x) \approx \log p_{\theta_\mathrm{base}}(x) + (\theta_\mathrm{toxic} - \theta_\mathrm{base})^\top \mathbf{\nabla_{\theta}}\log p_{\theta_\mathrm{base}}(x)
> $$
>
> Then, the contrastive score $s(x)$ can be rewritten as:
>
> $$
> s(x) = \log p_{\theta_\mathrm{base}}(x) - \log p_{\theta_\mathrm{toxic}}(x)
> \approx (\theta_\mathrm{base} - \theta_\mathrm{toxic})^\top \mathbf{\nabla_{\theta}}\log p_{\theta_\mathrm{base}}(x)
> $$
>
> ---
>
> # Additional Metrics and Comparisons with Recent Methods
>
> > **Comment:** The experimental setup can be improved to convince the community about the validity of the proposed method. I suggest the authors include some zero/few-shot metrics (e.g., MMLU) to understand how the model abilities are impacted.
> >
> > **Comment:** The evaluation proposed (methods compared) and metrics (PPL, n-gram) are too weak given the current abilities of LLMs and the thorough evaluations many works provide nowadays.
> >
> > **Comment:** In Section 3.1, I encourage the authors to include some metric related to zero/few-shot abilities. For example, the overall score in MMLU seems a good candidate.
> >
> > **Comment:** However, the universality claims are only validated using a small number of models and a non-comprehensive set of metrics.
>
> To address these concerns, we have incorporated **Few-Shot Accuracy** on the MMLU benchmark as an additional evaluation metric. (The results are shown below)
>
> > **Comment:** Results in Table 1. I have some concerns about the methods used for comparison. Although DExperts was a successful method when it was proposed, several methods have shown superior performance in recent years. For example, [SuauICML24] proposes a method to reduce toxicity without any inference cost and not requiring fine-tuning, showing superior performance compared to DExperts or pre-prompting. Similarly, [PozzobonEMNLP23] propose an efficient method for toxicity mitigation. I encourage the authors to consider comparing with some newer method than DExperts, this work's imact could strongly benefit from that. I also suggest adding these recent methods in the Related Work.
>
> We acknowledge the importance of comparing UniDetox with a broader range of baseline methods.
> In response to your suggestion, we have cited the works by Suau et al. [1] and Pozzobon et al. [2] in Section 4 of our revised manuscript.
>
> However, there are practical challenges in directly implementing these methods within the limited time of the author response period.
> AURA (Suau et al.)[1] requires the preparation of non-toxic data, which presents a challenge for a fair comparison with UniDetox and other baselines like Task Arithmetic and DExperts, which operate using toxic-only data.
> Goodtriever (Pozzobon et al.)[2] requires constructing specific datastores for each model with a different architecture, adding significant time and computational complexity. This was infeasible given the tight constraints of the response period.
>
> To strengthen our comparisons, we have incorporated **LM-Steer** [3], a recent method introduced in ACL 2024, which has demonstrated superior performance compared to DExperts.
> **LM-Steer** edits word embeddings during decoding time to steer a model's output.
> This method introduces a toxicity-steering matrix $W_\text{toxic}$ which is learned by fine-tuning on toxic data while keeping all other model parameters fixed.
> During decoding, detoxification is achieved by applying a linear perturbation to the word embedding using $W_\text{toxic}$:
>
> $$
> \mathbf{e}'(x_t) = \mathbf{e}(x_t) - \epsilon W_\text{toxic} \mathbf{e}(x_t)
> $$
>
> where $\mathbf{e}(x_t)$ denotes the word embedding of a token $x_t$, and $\epsilon$ is the hyperparameter controlling detoxification strength.
>
> Table 1+2 shows the updated experimental results, including LM-Steer and few-shot accuracy on MMLU.
> **As demonstrated, UniDetox consistently achieves the best detoxification results with a minor impact on downstream task performance**. For a complete evaluation, including diversity metrics, please refer to Section 3.5 in our revised paper.
>
> ---

---

> ### Author Response · Authors · 2024-11-21
> **Response to Reviewer 7CV4 2/4**
>
> **As demonstrated, UniDetox consistently achieves the best detoxification results with a minor impact on downstream task performance**. For a complete evaluation, including diversity metrics, please refer to Section 3.5 in our revised paper.
>
> ### Table1+2. Detoxification Results with an Additional Baseline LM-Steer and an Addition Metric MMLU Accuracy
>
> | Model                     | TP (ID) | TP (OOD) | EMT (ID) | EMT (OOD) | PPL   | MMLU (%) |
> |---------------------------|---------|----------|----------|-----------|-------|------------|
> | **GPT-2 XL**              | 0.53    | 0.41     | 0.54     | 0.43      | 17.28 | 32.07      |
> | PrePrompt_Short           | 0.58    | 0.49     | 0.56     | 0.49      | 23.61 | 31.87      |
> | PrePrompt_Long            | 0.63    | 0.53     | 0.61     | 0.54      | 13.51 | 30.31      |
> | Samples_GPT-2             | 0.48    | 0.35     | 0.49     | 0.38      | 15.71 | 32.20      |
> | LM-Steer                  | 0.44    | _0.32_   | 0.45     | 0.36      | 18.73 | 29.72      |
> | DExperts              | 0.50    | _0.35_   | 0.50     | 0.39      | 18.12 | 30.83      |
> | Task Arithmetic           | 0.52    | 0.38     | 0.52     | 0.40      | 17.64 | 29.92      |
> | UniDetox_GPT-2 (lr=5e-5) | **0.36** | **0.28** | **0.37** | **0.32** | 12.23 | 30.37      |
> | UniDetox_GPT-2 (lr=1e-5)  | _0.43_  | _0.32_   | _0.44_   | _0.35_    | 15.84 | 31.81      |
>
> | Model                     | TP (ID) | TP (OOD) | EMT (ID) | EMT (OOD) | PPL   | MMLU (%) |
> |---------------------------|---------|----------|----------|-----------|-------|------------|
> | **OPT-6.7B**              | 0.78    | 0.82     | 0.76     | 0.79      | 17.30 | 34.36      |
> | PrePrompt_Short           | 0.67    | 0.67     | 0.65     | 0.64      | 20.70 | 33.51      |
> | PrePrompt_Long            | 0.73    | 0.74     | 0.71     | 0.71      | 12.35 | 32.59      |
> | Samples_GPT-2             | 0.61    | 0.59     | 0.60     | 0.58      | 21.37 | 34.16      |
> | LM-Steer                  | 0.74    | 0.78     | 0.72     | 0.74      | 24.69 | 30.83      |
> | DExperts              | _0.62_  | _0.65_   | _0.60_   | _0.62_    | 28.19 | 35.40      |
> | Task Arithmetic           | 0.58    | 0.56     | 0.56     | 0.56      | 25.89 | 30.70      |
> | UniDetox_GPT-2 (lr=5e-5) | **0.20** | **0.14** | **0.23** | **0.19** | 13.57 | 31.16      |
> | UniDetox_GPT-2 (lr=1e-5)  | _0.50_  | _0.45_   | _0.50_   | _0.47_    | 25.82 | 32.27      |
>
> | Model                     | TP (ID) | TP (OOD) | EMT (ID) | EMT (OOD) | PPL   | MMLU (%) |
> |---------------------------|---------|----------|----------|-----------|-------|------------|
> | **Falcon-7B**             | 0.60    | 0.53     | 0.59     | 0.53      | 10.69 | 39.32      |
> | PrePrompt_Short           | 0.58    | 0.57     | 0.57     | 0.55      | 17.05 | 38.28      |
> | PrePrompt_Long            | 0.59    | 0.57     | 0.58     | 0.54      | 11.83 | 37.17      |
> | Samples_GPT-2             | 0.46    | 0.40     | 0.47     | 0.43      | 17.15 | 34.49      |
> | LM-Steer                  | 0.37    | 0.32     | 0.39     | 0.35      | 29.05 | 34.75      |
> | DExperts              | _0.30_  | 0.25     | _0.33_   | 0.28      | 28.71 | 37.88      |
> | Task Arithmetic           | 0.52    | 0.47     | 0.51     | 0.46      | 32.71 | 29.85      |
> | UniDetox_GPT-2 (lr=5e-5) | **0.26** | **0.22** | **0.29** | **0.26** | 8.78  | 34.23      |
> | UniDetox_GPT-2 (lr=1e-5)  | _0.30_  | _0.23_   | _0.32_   | _0.27_    | 29.54 | 38.28      |
>
> | Model                     | TP (ID) | TP (OOD) | EMT (ID) | EMT (OOD) | PPL   | MMLU (%) |
> |---------------------------|---------|----------|----------|-----------|-------|------------|
> | **LLaMA2-7B**             | 0.58    | 0.49     | 0.57     | 0.49      | 8.56  | 41.74      |
> | PrePrompt_Short           | 0.60    | 0.55     | 0.58     | 0.54      | 15.62 | 42.00      |
> | PrePrompt_Long            | 0.58    | 0.53     | 0.57     | 0.53      | 11.24 | 37.17      |
> | Samples_GPT-2             | 0.57    | 0.47     | 0.56     | 0.48      | 8.37  | 37.75      |
> | LM-Steer                  | 0.47    | 0.40     | 0.46     | 0.42      | 10.18 | 40.82      |
> | DExperts              | _0.45_  | _0.35_   | _0.44_   | _0.39_    | 9.91  | 39.71      |
> | Task Arithmetic           | 0.58    | 0.47     | 0.56     | 0.48      | 9.39  | 41.02      |
> | UniDetox_GPT-2 (lr=5e-5) | **0.20** | **0.16** | **0.25** | **0.20** | 9.44  | 36.25      |
> | UniDetox_GPT-2 (lr=1e-5)  | 0.50    | 0.37     | 0.49     | 0.40      | 9.19  | 38.28      |
>
> ---
>
> # Hyperparameter
>
> > **Comment:** Ablations: The proposed method has an important hyper-parameter $\lambda$. I missed an ablation study on $\lambda$, showing the impact of this choice in the final performance.
> >
> > **Comment:** Moreover, details about the choice of $\lambda$ and the data produced are lacking.
>
> We assume that the reviewer refers to the hyperparameter $\alpha$ as $\lambda$.
>
> In Table 5 (Appendix A.3), we evaluated the case when varying the value of  $\alpha$.

---

> ### Author Response · Authors · 2024-11-21
> **Response to Reviewer 7CV4 3/4**
>
> **Setting $\alpha$ to 0.05 still yields effective detoxification results, while the LM’s performance does not significantly change with the choice of $\alpha$**.
>  Thus, $\alpha$ does not have a significant impact on our results, and we consistently set  $\alpha=0.1$, aligning with prior works [4, 5, 6].
>
> ### Table 5. Detoxification Results for UniDetox with $\alpha=0.05$ and lr=1e-5
>
> | Model                             | TP (ID) | TP (OOD) | EMT (ID) | EMT (OOD) | PPL   | MMLU (%) |
> |-----------------------------------|---------|----------|----------|-----------|-------|----------|
> | **GPT-2 XL**                      | 0.53    | 0.41     | 0.54     | 0.43      | 17.28 | 32.07    |
> | UniDetox_GPT-2 ($\alpha$=0.05)    | 0.42    | 0.33     | 0.43     | 0.35      | 17.55 | 31.09    |
> | UniDetox_GPT-2 ($\alpha$=0.1)     | 0.43    | 0.32     | 0.44     | 0.35      | 15.84 | 31.81    |
> | **OPT-6.7B**                      | 0.78    | 0.82     | 0.76     | 0.79      | 17.30 | 34.36    |
> | UniDetox_GPT-2 ($\alpha$=0.05)    | 0.52    | 0.52     | 0.52     | 0.54      | 25.27 | 34.29    |
> | UniDetox_GPT-2 ($\alpha$=0.1)     | 0.50    | 0.45     | 0.50     | 0.47      | 25.82 | 32.27    |
> | **Falcon-7B**                     | 0.60    | 0.53     | 0.59     | 0.53      | 10.69 | 39.32    |
> | UniDetox_GPT-2 ($\alpha$=0.05)    | 0.37    | 0.29     | 0.39     | 0.34      | 22.28 | 35.40    |
> | UniDetox_GPT-2 ($\alpha$=0.1)     | 0.30    | 0.23     | 0.32     | 0.27      | 29.54 | 34.49    |
> | **LLaMA2-7B**                     | 0.58    | 0.49     | 0.57     | 0.49      | 8.56  | 41.74    |
> | UniDetox_GPT-2 ($\alpha$=0.05)    | 0.51    | 0.39     | 0.51     | 0.42      | 10.33 | 38.41    |
> | UniDetox_GPT-2 ($\alpha$=0.1)     | 0.50    | 0.37     | 0.49     | 0.40      | 9.19  | 38.28    |
>
> ---
> # Concerns about Perplexity
>
> > **Question:** Results in Table 1. Could the authors comment on the fact that PPL goes down from 17.28 to 12.23 when using UniDetox? To me, this behavior is counterintuitive, since the PPL of a LLM rarely goes down when one intervenes on the LLM. In this case, fine-tuning on distilled data is an intervention on the original model. It is surprising that fine-tuning on a small synthetic dataset brings the LLM perplexity to 12.23 (which would be the PPL of a much larger/stronger LLM). I encourage the authors to provide details about the PPL evaluation, and a conclusive justification of why such PPLs are strongly decreasing.
> >
> > **Comment:** I find very confusing the fact that PPL strongly decreases when the intervention is applied.
>
> We have detailed the calculation methods of perplexity in Appendix A.2 in our revised paper.
>
> We understand your concern regarding the drop in perplexity (PPL) observed when applying UniDetox. This is due to the adaptive plausibility constraint used for UniDetox. UniDetox generates detoxifying text using the plausibility constraint, which prevents the LMs from generating tokens with low probabilities. Thus, the detoxifying text is unlikely to contain tokens with low probabilities. By fine-tuning LMs on such detoxifying text, The fine-tuned LM becomes less likely to generate implausible tokens.
>
> As PPL is not sometimes sufficient for evaluating LM’s capabilities, we added the MMLU as an additional evaluation benchmark. We appreciate your suggestion to include more metrics, which has indeed provided a more comprehensive evaluation of our method.
>
> ---
>
> # OOD vs. ID Toxicity Reduction
>
> > **Question:** Results in Table 1. Why is OOD toxicity reduction stronger than ID reduction? I would have expected a fine-tuning based approach like UniDetox to be much more effective on ID. Could the authors comment on this aspect?
>
>  Such OOD generalization happens because a model that forgets words used to attack some social groups will also forget similar offensive words directed at other groups. Although the following is merely a hypothesis, our dataset distillation-based method can be interpreted as condensing toxic text across multiple domains and generating detoxifying text that makes the model unlearn this condensed toxicity. Through condensation, the detoxifying text captures more generalized notions of toxicity, which are beneficial for mitigating toxicity in OOD domains.
>
> ---
>
> # Evaluating Instruction-Tuned Models
>
> > **Question:** Results in Table 1. For the safety preprompt to be effective, have the authors considered evaluating an instruction-tuned model such as gemma-2-2b-it?
>
> Thank you for the suggestions. Following previous studies, we used models that are not instruction-tuned.
> Due to time and resource constraints during the response period, we have not yet evaluated UniDetox on instruction-tuned models like gemma-2-2b-it.
> However, we recognize the importance of this evaluation and plan to include it in our future work.

---

> ### Author Response · Authors · 2024-11-21
> **Response to Reviewer 7CV4 4/4**
>
> # Quantity and Examples of Distilled Text
>
> > **Comment:** The amount of data produced via distillation is of great importance. Can the authors discuss about the data required for UniDetox to be effective? Additionally, I would be really curious in seeing some of the distilled sentences produced by UniDetox
>
> As a detoxifying text, we sampled 640 texts, each with a maximum length of 256 tokens. We showed the details of hyperparameters in Appendix A.3.
> We also have included examples of detoxifying text in Appendix B.3, as well as sentences generated by models detoxified vias UniDetox in Appendix B.4 of our revised version.
>
> Although we do not yet fully understand the specific linguistic characteristics of detoxifying text that drive its effectiveness, our statistical analyses provide useful insights:
> - **Toxicity Analysis (Table 3, Sec. 3.6):** Detoxifying text shows consistently lower toxicity scores.
> - **Jaccard Similarity (Table 8, Appendix B.1):** Detoxifying text exhibits less overlap with DGHS toxic data, indicating a distinct distribution.
>
> These findings suggest that detoxifying text reduces toxic content and differs significantly from toxic data, enabling it to effectively guide models toward detoxification during fine-tuning.
>
> ---
>
> # Conclusion
> In summary, we have: **1)** included LM-Steer [3] as a new baseline to compare with UniDetox, **2)** incorporated Few-Shot Accuracy on the MMLU benchmark to provide a more comprehensive evaluation, **3)** presented results with a different hyperparameter configuration $\alpha = 0.05$ to showcase the robustness of UniDetox, and **4)** added examples of the distilled detoxifying text as well as the detoxified model generations.
>
> In the future, before camera-ready, we promise to **extend the method to instruction-fine-tuned models**.
>
> We hope that these revisions effectively address your concerns and demonstrate the efficacy of UniDetox.
> We believe that our approach offers a significant contribution to the field by providing a universally applicable detoxification method for large language models.
> We kindly request you to reconsider the evaluation score in light of these clarifications and additions.
>
> Thank you once again for your insightful feedback.
>
> Sincerely,
> The Authors
>
> ---
>
> **References:**
>
> [1] Whispering Experts: Neural Interventions for Toxicity Mitigation in Language Models. Suau et al. ICML 2024
>
> [2] Goodtriever: Adaptive Toxicity Mitigation with Retrieval-augmented Models. Pozzobon et al. EMNLP 2023
>
> [3] Word Embeddings Are Steers for Language Models. Han et al.ACL 2024
>
> [4] DExperts: Decoding-Time Controlled Text Generation with Experts and Anti-Experts. Liu et al. ACL 2021
>
> [5] Contrastive Decoding: Open-ended Text Generation as Optimization. Li et al. ACL 2023
>
> [6] Contrastive Decoding Improves Reasoning in Large Language Models. O'Brien and Lewis. arXiv Preprint 2309.09117, 2023

---

> ### Author Response · Authors · 2024-11-25
> **Response to Reviewer 7CV4: Kind Reminder**
>
> Dear Reviewer 7CV4,
>
> Thank you for your thoughtful feedback, which has played a valuable role in improving our paper. As we near the end of the discussion period, we hope that our responses have adequately addressed the concerns you raised.
>
> We would greatly appreciate it if you could let us know whether our revisions and clarifications have resolved your concerns. If there are any remaining questions or areas that need further clarification, please let us know.
>
> Thank you again for your time and consideration.
>
> Sincerely,
>
> The Authors

---

> > ### Author Response · Authors · 2024-11-28
> >
> > Dear Reviewer 7CV4,
> >
> > We sincerely apologize for following up with another reminder, but we wished to kindly bring to your attention that we have made substantial revisions to the paper, including new experiments aimed at addressing your concerns (as detailed in our previous messages).
> >
> > We would greatly appreciate your thoughts on whether these updates adequately address your critiques.
> >
> > Sincerely,
> >
> > The Authors

---

> > > ### Comment · Reviewer_7CV4 · 2024-11-28
> > >
> > > I want to first apologize for this late response. I could not write before for personal reasons, sorry for the inconveniences this might have caused and the _stress_ I know incurs in authors during the review.
> > >
> > > Thanks for the  thorough rebuttal:
> > >
> > > * Steps for Taylor expansion are clear. I encourage the authors to include them in the final paper.
> > > * I strongly appreciate the inclusion of an additional method for comparison (LM-Steer, ACL24). I also appreciate the authors including more references in the paper, as well as an explanation about the reasons why these are not included in the experiments.
> > > * The explanation about the drop in PPL with UnitDetox seems reasonable. I still think that a drop in PPL has some additional impact on the text generated (less diverse probably, due to the use of very likely tokens all along). However, the inclusion of MMLU makes me be more confident about the LLM abilities once intervened.
> > > * I accknowledge that including new models is hard to achieve during the rebuttal, but I appreciate the willingness to include instruction-tuned models in the final version.
> > >
> > > Accounting for all the above, I believe this work has increased in quality, thus I update my score to 6, above acceptance.

---

> > > > ### Author Response · Authors · 2024-11-28
> > > >
> > > > Dear Reviewer 7CV4,
> > > >
> > > > Thank you for taking the time to respond despite your personal circumstances. We truly appreciate your consideration and understanding.
> > > >
> > > > We are grateful that you took the time to read our responses and the revised paper. Your invaluable suggestions, especially about adding MMLU as an additional benchmark, have significantly enhanced the soundness of our study. We sincerely appreciate your input.
> > > >
> > > > Additionally, we will incorporate your recommendation to include the Taylor expansion in the final version of our paper, along with the instruction-fine-tuned models as previously promised.
> > > >
> > > > Thank you once again for your constructive feedback and support.
> > > >
> > > > Sincerely,
> > > >
> > > > The Authors

---

### Official Review · Reviewer_gfVU · 2024-11-01

**Soundness:** 2
**Presentation:** 2
**Contribution:** 2
**Rating:** 5
**Confidence:** 3

**Summary:**

UniDetox proposes a framework for the detoxification of Language Models via a three-step process.
* Step 1 : Create a Toxic Model
* Step 2 : Distill Detoxifying Text
* Step 3 : Finetune on Detoxifying Text

It is valuable in that it proposes a way to detoxify LLMs in an efficient manner and also show that the steps till Step 2, could be independent of the actual model.
This is very favorable as models are getting larger and detoxifying them with common methods like task vectors can get computationally expensive (since they involve training another model of identical specs)

**Strengths:**

* Definitely an interesting idea combining the core principles of task arithmetic and dataset distillation (i.e  showing that fine-tuning on data from a specific distribution also pivots a model’s representation internally in that direction and applying it to the toxicity setting)
* The idea that detoxifying text can be independent of the final model and can come from a smaller model is also favorable
* Their baseline approach also contains the straightforward “prompt” variants which is a useful data point for comparison as models get more powerful.

**Weaknesses:**

I would like to list the following weakness fully ensuring the authors that I am not unreasonable and am completely open to increasing my score if these are addressed/answered satisfactorily.

* The related section definitely lacks coverage of methods that are available for detoxification. Few of them include : Parameter-Efficient Detoxification with Contrastive Decoding, etc
* There is a lack in comparison with more recent methods. An efficient method to benchmark against would be the cited LM Steering work(Han et al)
* Most work in Detoxification uses the Perspective API in their evaluations, but it has been omitted here without any explanation.
* An example of what the generated“detoxifying” text looks like would help. Is this just “non-toxic” text ? Or does it have other interesting lexical properties? Some more analysis would be welcome here( even if in the appendix)

**Questions:**

* Very curious to know why recent methods weren’t benchmarked against?
* Is there an obvious reason the Perspective API wasn’t used ?

---

> ### Author Response · Authors · 2024-11-21
> **Response to Reviewer gfVU 1/3**
>
> Dear Reviewer gfVU,
>
> Thank you for your valuable feedback on our submission.
> We greatly appreciate the time and effort you have taken to review our work and provide constructive insights.
> Below, we address your concerns in detail.
>
> ---
>
> # Related Work Coverage
>
> > **Comment:** The related section definitely lacks coverage of methods that are available for detoxification. Few of them include: Parameter-Efficient Detoxification with Contrastive Decoding, etc.
>
> In our revised paper, we have expanded the related work section to include five additional methods, such as Parameter-Efficient Detoxification with Contrastive Decoding [1], Ethos [2], ProFS [3], AURA [4], and Goodtriever [5].
>
> ---
>
> # Comparisons with Recent Methods
>
> > **Comment:** There is a lack in comparison with more recent methods. An efficient method to benchmark against would be the cited LM Steering work(Han et al)
> >
> > **Question:** Very curious to know why recent methods weren’t benchmarked against?
>
> To strengthen our comparisons, we have incorporated **LM-Steer** [6], a recent method introduced in ACL 2024, which has demonstrated superior performance compared to DExperts.
>
> We have also introduced **Few-Shot Accuracy** on the MMLU benchmark to ensure that the detoxified LMs maintain the performance on several tasks.
>
> Table 1+2 shows the updated experimental results, including LM-Steer and few-shot accuracy on MMLU.
> **As demonstrated, UniDetox consistently achieves the best detoxification results with a minor impact on downstream task performance**.
>  For a complete evaluation, including diversity metrics, please refer to Section 3.5 in our revised paper.
>
> ---
>
> # Use of Perspective API
>
> > **Comment:** Most work in Detoxification uses the Perspective API in their evaluations, but it has been omitted here without any explanation.
> >
> > **Question:** Is there an obvious reason the Perspective API wasn’t used?
>
> We did not use the Perspective API due to its stringent query-per-second (QPS) limit of 1 query per second, which poses a significant challenge for our evaluation setup. (https://developers.perspectiveapi.com/s/about-the-api-limits-and-errors?language=en_US)
> With nearly 30,000 entries per model and 20 models per baseline for hyperparameter tuning, this limitation renders the Perspective API infeasible for our experiments.
>
> However, we have reached out to the Perspective API team to request an increased QPS.
> If granted, we commit to including Perspective API evaluations in the Appendix for the camera-ready version.
>
> —
>
> # Examples of Detoxifying Text and Analysis
>
> > **Comment:** An example of what the generated“detoxifying” text looks like would help. Is this just “non-toxic” text ? Or does it have other interesting lexical properties? Some more analysis would be welcome here( even if in the appendix)
>
> We appreciate this suggestion and have included examples of detoxifying text in Appendix B.3 of our revised paper.
>
> Although we do not yet fully understand the specific linguistic characteristics of detoxifying text that drive its effectiveness, our statistical analyses provide useful insights:
> - **Toxicity Analysis (Table 3, Sec. 3.6):** Detoxifying text shows consistently lower toxicity scores.
> - **Jaccard Similarity (Table 8, Appendix B.1):** Detoxifying text exhibits less overlap with DGHS toxic data, indicating a distinct distribution.
>
> These findings suggest that detoxifying text reduces toxic content and differs significantly from toxic data, enabling it to effectively guide models toward detoxification during fine-tuning.
>
> ---

---

> ### Author Response · Authors · 2024-11-21
> **Response to Reviewer gfVU 2/3**
>
> **As demonstrated, UniDetox consistently achieves the best detoxification results with a minor impact on downstream task performance**. For a complete evaluation, including diversity metrics, please refer to Section 3.5 in our revised paper.
>
> ### Table1+2. Detoxification Results with an Additional Baseline LM-Steer and an Addition Metric MMLU Accuracy
>
> | Model                     | TP (ID) | TP (OOD) | EMT (ID) | EMT (OOD) | PPL   | MMLU (%) |
> |---------------------------|---------|----------|----------|-----------|-------|------------|
> | **GPT-2 XL**              | 0.53    | 0.41     | 0.54     | 0.43      | 17.28 | 32.07      |
> | PrePrompt_Short           | 0.58    | 0.49     | 0.56     | 0.49      | 23.61 | 31.87      |
> | PrePrompt_Long            | 0.63    | 0.53     | 0.61     | 0.54      | 13.51 | 30.31      |
> | Samples_GPT-2             | 0.48    | 0.35     | 0.49     | 0.38      | 15.71 | 32.20      |
> | LM-Steer                  | 0.44    | _0.32_   | 0.45     | 0.36      | 18.73 | 29.72      |
> | DExperts              | 0.50    | _0.35_   | 0.50     | 0.39      | 18.12 | 30.83      |
> | Task Arithmetic           | 0.52    | 0.38     | 0.52     | 0.40      | 17.64 | 29.92      |
> | UniDetox_GPT-2 (lr=5e-5) | **0.36** | **0.28** | **0.37** | **0.32** | 12.23 | 30.37      |
> | UniDetox_GPT-2 (lr=1e-5)  | _0.43_  | _0.32_   | _0.44_   | _0.35_    | 15.84 | 31.81      |
>
> | Model                     | TP (ID) | TP (OOD) | EMT (ID) | EMT (OOD) | PPL   | MMLU (%) |
> |---------------------------|---------|----------|----------|-----------|-------|------------|
> | **OPT-6.7B**              | 0.78    | 0.82     | 0.76     | 0.79      | 17.30 | 34.36      |
> | PrePrompt_Short           | 0.67    | 0.67     | 0.65     | 0.64      | 20.70 | 33.51      |
> | PrePrompt_Long            | 0.73    | 0.74     | 0.71     | 0.71      | 12.35 | 32.59      |
> | Samples_GPT-2             | 0.61    | 0.59     | 0.60     | 0.58      | 21.37 | 34.16      |
> | LM-Steer                  | 0.74    | 0.78     | 0.72     | 0.74      | 24.69 | 30.83      |
> | DExperts              | _0.62_  | _0.65_   | _0.60_   | _0.62_    | 28.19 | 35.40      |
> | Task Arithmetic           | 0.58    | 0.56     | 0.56     | 0.56      | 25.89 | 30.70      |
> | UniDetox_GPT-2 (lr=5e-5) | **0.20** | **0.14** | **0.23** | **0.19** | 13.57 | 31.16      |
> | UniDetox_GPT-2 (lr=1e-5)  | _0.50_  | _0.45_   | _0.50_   | _0.47_    | 25.82 | 32.27      |
>
> | Model                     | TP (ID) | TP (OOD) | EMT (ID) | EMT (OOD) | PPL   | MMLU (%) |
> |---------------------------|---------|----------|----------|-----------|-------|------------|
> | **Falcon-7B**             | 0.60    | 0.53     | 0.59     | 0.53      | 10.69 | 39.32      |
> | PrePrompt_Short           | 0.58    | 0.57     | 0.57     | 0.55      | 17.05 | 38.28      |
> | PrePrompt_Long            | 0.59    | 0.57     | 0.58     | 0.54      | 11.83 | 37.17      |
> | Samples_GPT-2             | 0.46    | 0.40     | 0.47     | 0.43      | 17.15 | 34.49      |
> | LM-Steer                  | 0.37    | 0.32     | 0.39     | 0.35      | 29.05 | 34.75      |
> | DExperts              | _0.30_  | 0.25     | _0.33_   | 0.28      | 28.71 | 37.88      |
> | Task Arithmetic           | 0.52    | 0.47     | 0.51     | 0.46      | 32.71 | 29.85      |
> | UniDetox_GPT-2 (lr=5e-5) | **0.26** | **0.22** | **0.29** | **0.26** | 8.78  | 34.23      |
> | UniDetox_GPT-2 (lr=1e-5)  | _0.30_  | _0.23_   | _0.32_   | _0.27_    | 29.54 | 38.28      |
>
> | Model                     | TP (ID) | TP (OOD) | EMT (ID) | EMT (OOD) | PPL   | MMLU (%) |
> |---------------------------|---------|----------|----------|-----------|-------|------------|
> | **LLaMA2-7B**             | 0.58    | 0.49     | 0.57     | 0.49      | 8.56  | 41.74      |
> | PrePrompt_Short           | 0.60    | 0.55     | 0.58     | 0.54      | 15.62 | 42.00      |
> | PrePrompt_Long            | 0.58    | 0.53     | 0.57     | 0.53      | 11.24 | 37.17      |
> | Samples_GPT-2             | 0.57    | 0.47     | 0.56     | 0.48      | 8.37  | 37.75      |
> | LM-Steer                  | 0.47    | 0.40     | 0.46     | 0.42      | 10.18 | 40.82      |
> | DExperts              | _0.45_  | _0.35_   | _0.44_   | _0.39_    | 9.91  | 39.71      |
> | Task Arithmetic           | 0.58    | 0.47     | 0.56     | 0.48      | 9.39  | 41.02      |
> | UniDetox_GPT-2 (lr=5e-5) | **0.20** | **0.16** | **0.25** | **0.20** | 9.44  | 36.25      |
> | UniDetox_GPT-2 (lr=1e-5)  | 0.50    | 0.37     | 0.49     | 0.40      | 9.19  | 38.28      |

---

> ### Author Response · Authors · 2024-11-21
> **Response to Reviwer gfVU 3/3**
>
> # Conclusion
> In summary, we have: **1)** included LM-Steer [3] as a new baseline to compare with UniDetox, **2)** incorporated Few-Shot Accuracy on the MMLU benchmark to provide a more comprehensive evaluation, and **3)** added examples of the distilled detoxifying text.
>
> In the future, **before camera-ready, we commit to re-evaluating using Perspective API**.
>
> We hope that our responses effectively address your concerns and demonstrate the efficacy of UniDetox.
> We believe that our approach offers a significant contribution to the field by providing a universally applicable detoxification method for large language models.
> We kindly request you to reconsider the evaluation score in light of these clarifications and additions.
>
> Thank you once again for your insightful feedback.
>
> Sincerely,
> The Authors
>
> ---
>
> **References:**
>
> [1] Parameter-Efficient Detoxification with Contrastive Decoding. Niu et al. HuCLLM 2024
>
> [2] Ethos: Rectifying Language Models in Orthogonal Parameter Space. Gao et al. NAACL 2024
>
> [3] Model Editing as a Robust and Denoised variant of DPO: A Case Study on Toxicity. Uppaal et al. arXiv Preprint 2405.13967, 2024
>
> [4] Whispering Experts: Neural Interventions for Toxicity Mitigation in Language Models. Suau et al. ICML 2024
>
> [5] Goodtriever: Adaptive Toxicity Mitigation with Retrieval-augmented Models. Pozzobon et al. EMNLP 2023
>
> [6] Word Embeddings Are Steers for Language Models. Han et al.ACL 2024

---

> > ### Comment · Reviewer_gfVU · 2024-11-25
> >
> > The authors have addressed my concerns. I will update my score accordingly.

---

> > > ### Author Response · Authors · 2024-11-25
> > > **Response to Reviewer gfVU**
> > >
> > > Dear Reviewer gfVU,
> > >
> > > Thank you for raising your score from 3 to 5—we truly appreciate your willingness to reconsider our work and are glad that we were able to address your concerns.
> > >
> > > We noticed that your current score is still marginally below the acceptance threshold. Since there is still time before the discussion period ends, we wanted to check if there are any remaining issues or questions that we haven't fully addressed. We would be more than happy to provide further clarification or additional information to alleviate any lingering concerns.
> > >
> > > We are grateful for your recognition of the novelty of our approach in combining the core principles of task arithmetic and dataset distillation. We genuinely believe that our paper could offer significant benefits to the community and would greatly appreciate the opportunity for it to reach a wider audience.
> > >
> > > Thank you once again for your thoughtful review and consideration.
> > >
> > > Sincerely,
> > >
> > > The Authors

---

### Official Review · Reviewer_jYcm · 2024-11-04

**Soundness:** 2
**Presentation:** 3
**Contribution:** 2
**Rating:** 6
**Confidence:** 4

**Summary:**

This paper introduces UniDetox, a universal detoxification method for LLMs that avoids model-specific tuning by generating detoxifying text via dataset distillation with contrastive decoding. This distilled text derived from GPT-2 can detoxify other models like LLaMA-2 without additional hyperparameter adjustments. Experiments validate UniDetox's ability to mitigate toxicity and reduce political bias consistently across models. The paper also discusses insights into the features necessary for detoxification.

**Strengths:**

- Proposes a universally applicable detoxification approach, addressing limitations of existing model-specific methods by utilizing dataset distillation.
- Provides a theoretical basis by aligning detoxification with the direction opposite to a "toxic vector" in model parameter space (Sec. 2.2, Eq. 5)
- Reduces the need for model-specific hyperparameter tuning, as evidenced by successful application to different models with consistent results with a single hyperparameter configuration.

**Weaknesses:**

- The paper lacks a comparison with alternative alignment algorithms such as DPO or other detoxification baselines using task arithmetic between a base model and a tuned model, such as [1][2].
- Although UNIDETOX is free from downstream model-specific hyperparameter tuning, results are sensitive to the selected learning rate and α parameters (Section 3.4). The need for different UNIDETOX configurations impacting performance lacks sufficient justification or analysis.
- UniDetox demonstrates a drop in diversity metrics compared to other methods. A broader evaluation on other LM tasks would help validate its capability to preserve language quality.

[1] https://arxiv.org/abs/2403.08994
[2] https://arxiv.org/abs/2405.13967

**Questions:**

- Would the authors provide examples of the distilled detoxifying text for tuning? This would help clarify the properties and coherence of the distilled data.

---

> ### Author Response · Authors · 2024-11-21
> **Response to Reviewer jYcm 1/3**
>
> Dear Reviewer jYcm,
>
> Thank you for your valuable feedback on our submission. We appreciate the time and effort you have invested in reviewing our work and providing constructive comments. Below, we address each of your concerns in detail.
>
> ---
>
> # Additional Baseline Methods
>
> > **Comment:** The paper lacks a comparison with alternative alignment algorithms such as DPO or other detoxification baselines using task arithmetic between a base model and a tuned model, such as [1][2].
>
> We acknowledge the importance of comparing UniDetox with a broader range of baseline methods. In response to your suggestion, we have cited the works by Gao et al. [1] and Uppaal et al. [2] in Section 4 of our revised manuscript. However, we observed that both Ethos [1] and ProFS [2] require the preparation of non-toxic data for implementation. This requirement presents a challenge for fair comparison with our approach and other baselines like Task Arithmetic and DExperts, which operate using toxic-only data.
>
> To strengthen our comparisons, we have incorporated **LM-Steer** [3], a recent method introduced in ACL 2024, which has demonstrated superior performance compared to DExperts. Below is a brief overview of LM-Steer:
>
> **LM-Steer** edits word embeddings during decoding time to steer a model's output.
> This method introduces a toxicity-steering matrix $W_\text{toxic}$ which is learned by fine-tuning on toxic data while keeping all other model parameters fixed.
> During decoding, detoxification is achieved by applying a linear perturbation to the word embedding using $W_\text{toxic}$:
>
> $$
> \mathbf{e}'(x_t) = \mathbf{e}(x_t) - \epsilon W_\text{toxic} \mathbf{e}(x_t)
> $$
>
> where $\mathbf{e}(x_t)$ denotes the word embedding of a token $x_t$, and $\epsilon$ is the hyperparameter controlling detoxification strength.
>
> ---
>
> # Additional Metrics
>
> > **Comment:** UniDetox demonstrates a drop in diversity metrics compared to other methods. A broader evaluation on other LM tasks would help validate its capability to preserve language quality.
>
> To address the concern regarding diversity metrics and to provide a more comprehensive evaluation, we have introduced **Few-Shot Accuracy** on the MMLU benchmark as an additional metric.
>
> Table1+2 shows the updated experimental results (the Table was shown below in response 2/3), including LM-Steer and few-shot accuracy on MMLU.
> **As demonstrated, UniDetox consistently achieves the best detoxification results with a minor impact on downstream task performance**. For a complete evaluation, including diversity metrics, please refer to Section 3.5 in our revised paper.
>
> # Hyperparameter Configuration
>
> > **Comment:** Although UniDetox is free from downstream model-specific hyperparameter tuning, results are sensitive to the selected learning rate and $\alpha$ parameters (Section 3.4). The need for different UniDetox configurations impacting performance lacks sufficient justification or analysis.
>
>
> In UniDetox, the primary hyperparameters are the learning rate and the adaptive plausibility constraint $\alpha$. As for the learning rate, in both cases of 5e-5 and 1e-5, Unidetox exhibits higher detoxifying performance compared to other methods while maintaining LM’s ability, as shown in Table 1 and 2.
>
> Regarding the adaptive plausibility constraint $\alpha$, we conducted a study when using different values of $\alpha$. We present detoxification performances using  $\alpha=0.05$ and $\alpha=0.1$ in Table 5 (the Table was shown below in response 3/3). **Setting $\alpha$ to 0.05 still yields effective detoxification results, while the LM’s performance does not significantly change with the choice of $\alpha$.** Thus, $\alpha$ does not have a significant impact on our results, and we consistently set  $\alpha=0.1$, aligning with prior works [4, 5, 6].
>
> For a complete evaluation, including diversity metrics, please refer to Appendix A.3 in our revised paper.
>
> ---
>
> # Examples of Distilled Detoxifying Text
>
> > **Question:** Would the authors provide examples of the distilled detoxifying text for tuning? This would help clarify the properties and coherence of the distilled data.
>
> We have included examples of the distilled detoxifying text in Appendix B.3 of our revised manuscript.
>
> Although we do not yet fully understand the specific linguistic characteristics of detoxifying text that drive its effectiveness, our statistical analyses provide useful insights:
> - **Toxicity Analysis (Table 3, Sec. 3.6):** Detoxifying text shows consistently lower toxicity scores.
> - **Jaccard Similarity (Table 8, Appendix B.1):** Detoxifying text exhibits less overlap with DGHS toxic data, indicating a distinct distribution.
>
> These findings suggest that detoxifying text reduces toxic content and differs significantly from toxic data, enabling it to effectively guide models toward detoxification during fine-tuning.

---

> ### Author Response · Authors · 2024-11-21
> **Response to Reviewer jYcm 2/3**
>
> **As demonstrated, UniDetox consistently achieves the best detoxification results with a minor impact on downstream task performance**. For a complete evaluation, including diversity metrics, please refer to Section 3.5 in our revised paper.
>
> ### Table1+2. Detoxification Results with an Additional Baseline LM-Steer and an Addition Metric MMLU Accuracy
>
> | Model                     | TP (ID) | TP (OOD) | EMT (ID) | EMT (OOD) | PPL   | MMLU (%) |
> |---------------------------|---------|----------|----------|-----------|-------|------------|
> | **GPT-2 XL**              | 0.53    | 0.41     | 0.54     | 0.43      | 17.28 | 32.07      |
> | PrePrompt_Short           | 0.58    | 0.49     | 0.56     | 0.49      | 23.61 | 31.87      |
> | PrePrompt_Long            | 0.63    | 0.53     | 0.61     | 0.54      | 13.51 | 30.31      |
> | Samples_GPT-2             | 0.48    | 0.35     | 0.49     | 0.38      | 15.71 | 32.20      |
> | LM-Steer                  | 0.44    | _0.32_   | 0.45     | 0.36      | 18.73 | 29.72      |
> | DExperts              | 0.50    | _0.35_   | 0.50     | 0.39      | 18.12 | 30.83      |
> | Task Arithmetic           | 0.52    | 0.38     | 0.52     | 0.40      | 17.64 | 29.92      |
> | UniDetox_GPT-2 (lr=5e-5) | **0.36** | **0.28** | **0.37** | **0.32** | 12.23 | 30.37      |
> | UniDetox_GPT-2 (lr=1e-5)  | _0.43_  | _0.32_   | _0.44_   | _0.35_    | 15.84 | 31.81      |
>
> | Model                     | TP (ID) | TP (OOD) | EMT (ID) | EMT (OOD) | PPL   | MMLU (%) |
> |---------------------------|---------|----------|----------|-----------|-------|------------|
> | **OPT-6.7B**              | 0.78    | 0.82     | 0.76     | 0.79      | 17.30 | 34.36      |
> | PrePrompt_Short           | 0.67    | 0.67     | 0.65     | 0.64      | 20.70 | 33.51      |
> | PrePrompt_Long            | 0.73    | 0.74     | 0.71     | 0.71      | 12.35 | 32.59      |
> | Samples_GPT-2             | 0.61    | 0.59     | 0.60     | 0.58      | 21.37 | 34.16      |
> | LM-Steer                  | 0.74    | 0.78     | 0.72     | 0.74      | 24.69 | 30.83      |
> | DExperts              | _0.62_  | _0.65_   | _0.60_   | _0.62_    | 28.19 | 35.40      |
> | Task Arithmetic           | 0.58    | 0.56     | 0.56     | 0.56      | 25.89 | 30.70      |
> | UniDetox_GPT-2 (lr=5e-5) | **0.20** | **0.14** | **0.23** | **0.19** | 13.57 | 31.16      |
> | UniDetox_GPT-2 (lr=1e-5)  | _0.50_  | _0.45_   | _0.50_   | _0.47_    | 25.82 | 32.27      |
>
> | Model                     | TP (ID) | TP (OOD) | EMT (ID) | EMT (OOD) | PPL   | MMLU (%) |
> |---------------------------|---------|----------|----------|-----------|-------|------------|
> | **Falcon-7B**             | 0.60    | 0.53     | 0.59     | 0.53      | 10.69 | 39.32      |
> | PrePrompt_Short           | 0.58    | 0.57     | 0.57     | 0.55      | 17.05 | 38.28      |
> | PrePrompt_Long            | 0.59    | 0.57     | 0.58     | 0.54      | 11.83 | 37.17      |
> | Samples_GPT-2             | 0.46    | 0.40     | 0.47     | 0.43      | 17.15 | 34.49      |
> | LM-Steer                  | 0.37    | 0.32     | 0.39     | 0.35      | 29.05 | 34.75      |
> | DExperts              | _0.30_  | 0.25     | _0.33_   | 0.28      | 28.71 | 37.88      |
> | Task Arithmetic           | 0.52    | 0.47     | 0.51     | 0.46      | 32.71 | 29.85      |
> | UniDetox_GPT-2 (lr=5e-5) | **0.26** | **0.22** | **0.29** | **0.26** | 8.78  | 34.23      |
> | UniDetox_GPT-2 (lr=1e-5)  | _0.30_  | _0.23_   | _0.32_   | _0.27_    | 29.54 | 38.28      |
>
> | Model                     | TP (ID) | TP (OOD) | EMT (ID) | EMT (OOD) | PPL   | MMLU (%) |
> |---------------------------|---------|----------|----------|-----------|-------|------------|
> | **LLaMA2-7B**             | 0.58    | 0.49     | 0.57     | 0.49      | 8.56  | 41.74      |
> | PrePrompt_Short           | 0.60    | 0.55     | 0.58     | 0.54      | 15.62 | 42.00      |
> | PrePrompt_Long            | 0.58    | 0.53     | 0.57     | 0.53      | 11.24 | 37.17      |
> | Samples_GPT-2             | 0.57    | 0.47     | 0.56     | 0.48      | 8.37  | 37.75      |
> | LM-Steer                  | 0.47    | 0.40     | 0.46     | 0.42      | 10.18 | 40.82      |
> | DExperts              | _0.45_  | _0.35_   | _0.44_   | _0.39_    | 9.91  | 39.71      |
> | Task Arithmetic           | 0.58    | 0.47     | 0.56     | 0.48      | 9.39  | 41.02      |
> | UniDetox_GPT-2 (lr=5e-5) | **0.20** | **0.16** | **0.25** | **0.20** | 9.44  | 36.25      |
> | UniDetox_GPT-2 (lr=1e-5)  | 0.50    | 0.37     | 0.49     | 0.40      | 9.19  | 38.28      |

---

> ### Author Response · Authors · 2024-11-21
> **Response to Reviewer jYcm 3/3**
>
> **Setting $\alpha$ to 0.05 still yields effective detoxification results, while the LM’s performance does not significantly change with the choice of $\alpha$.** Thus, $\alpha$ does not have a significant impact on our results, and we consistently set  $\alpha=0.1$, aligning with prior works [4, 5, 6].
>
> For a complete evaluation, including diversity metrics, please refer to Appendix A.3 in our revised paper.
>
> ### Table 5. Detoxification Results for UniDetox with $\alpha=0.05$ and lr=1e-5
>
> | Model                             | TP (ID) | TP (OOD) | EMT (ID) | EMT (OOD) | PPL   | MMLU (%) |
> |-----------------------------------|---------|----------|----------|-----------|-------|----------|
> | **GPT-2 XL**                      | 0.53    | 0.41     | 0.54     | 0.43      | 17.28 | 32.07    |
> | UniDetox_GPT-2 ($\alpha$=0.05)    | 0.42    | 0.33     | 0.43     | 0.35      | 17.55 | 31.09    |
> | UniDetox_GPT-2 ($\alpha$=0.1)     | 0.43    | 0.32     | 0.44     | 0.35      | 15.84 | 31.81    |
> | **OPT-6.7B**                      | 0.78    | 0.82     | 0.76     | 0.79      | 17.30 | 34.36    |
> | UniDetox_GPT-2 ($\alpha$=0.05)    | 0.52    | 0.52     | 0.52     | 0.54      | 25.27 | 34.29    |
> | UniDetox_GPT-2 ($\alpha$=0.1)     | 0.50    | 0.45     | 0.50     | 0.47      | 25.82 | 32.27    |
> | **Falcon-7B**                     | 0.60    | 0.53     | 0.59     | 0.53      | 10.69 | 39.32    |
> | UniDetox_GPT-2 ($\alpha$=0.05)    | 0.37    | 0.29     | 0.39     | 0.34      | 22.28 | 35.40    |
> | UniDetox_GPT-2 ($\alpha$=0.1)     | 0.30    | 0.23     | 0.32     | 0.27      | 29.54 | 34.49    |
> | **LLaMA2-7B**                     | 0.58    | 0.49     | 0.57     | 0.49      | 8.56  | 41.74    |
> | UniDetox_GPT-2 ($\alpha$=0.05)    | 0.51    | 0.39     | 0.51     | 0.42      | 10.33 | 38.41    |
> | UniDetox_GPT-2 ($\alpha$=0.1)     | 0.50    | 0.37     | 0.49     | 0.40      | 9.19  | 38.28    |
>
> ---
>
> # Conclusion
> In summary, we have: **1)** included LM-Steer [3] as a new baseline to compare with UniDetox, **2)** incorporated Few-Shot Accuracy on the MMLU benchmark to provide a more comprehensive evaluation, **3)** presented results with a different hyperparameter configuration $\alpha = 0.05$ to showcase the robustness of UniDetox, and **4)** added examples of the distilled detoxifying text.
>
> We hope that our responses effectively address your concerns and demonstrate the efficacy of UniDetox.
> We believe that our approach offers a significant contribution to the field by providing a universally applicable detoxification method for large language models.
> We kindly request you to reconsider the evaluation score in light of these clarifications and additions.
>
> Thank you once again for your insightful feedback.
>
> Sincerely,
> The Authors
>
> ---
>
> **References:**
>
> [1] Ethos: Rectifying Language Models in Orthogonal Parameter Space. Gao et al. NAACL 2024
>
> [2] Model Editing as a Robust and Denoised variant of DPO: A Case Study on Toxicity. Uppaal et al. arXiv Preprint 2405.13967, 2024
>
> [3] Word Embeddings Are Steers for Language Models. Han et al.ACL 2024
>
> [4] DExperts: Decoding-Time Controlled Text Generation with Experts and Anti-Experts. Liu et al. ACL 2021
>
> [5] Contrastive Decoding: Open-ended Text Generation as Optimization. Li et al. ACL 2023
>
> [6] Contrastive Decoding Improves Reasoning in Large Language Models. O'Brien and Lewis. arXiv Preprint 2309.09117, 2023

---

> ### Author Response · Authors · 2024-11-25
> **Response to Reviewer jYcm: Kind Reminder**
>
> Dear Reviewer jYcm,
>
> Thank you for your thoughtful feedback, which has played a valuable role in improving our paper. As we near the end of the discussion period, we hope that our responses have adequately addressed the concerns you raised.
>
> We would greatly appreciate it if you could let us know whether our revisions and clarifications have resolved your concerns. If there are any remaining questions or areas that need further clarification, please let us know.
>
> Thank you again for your time and consideration.
>
> Sincerely,
>
> The Authors

---

> > ### Author Response · Authors · 2024-11-28
> >
> > Dear Reviewer jYcm,
> >
> > We sincerely apologize for following up with another reminder, but we wished to kindly bring to your attention that we have made substantial revisions to the paper, including new experiments aimed at addressing your concerns (as detailed in our previous messages).
> >
> > We would greatly appreciate your thoughts on whether these updates adequately address your critiques.
> >
> > Sincerely,
> >
> > The Authors

---

> ### Author Response · Authors · 2024-12-03
>
> Dear Reviewer jYcm,
>
> This is a final reminder that today is the last day the reviewers can post messages on the forum. We understand it’s a busy time, but we’d greatly appreciate hearing your thoughts on the revised paper and responses.
>
> We’ve made significant updates, including new experiments, to address your and other reviewers’ concerns. The other reviewers have shared positive feedback on the revisions, and your feedback would also be invaluable.
>
> Thank you for your time,
>
> Sincerely,
>
> The Authors

---

### Author Response · Authors · 2024-12-02
**Global Response: Thank You to All Reviewers**

We sincerely thank all the reviewers for their time and thoughtful feedback. We are grateful for your recognition of our work as novel and interesting (gfVU, 7CV4, zPHB), effective and efficient (jYcm, gfVU, zPHB), and clear and easy to follow (7CV4, zPHB).
We also appreciate your acknowledgment that our work reduces the need for model-specific implementation and hyperparameter tuning with a theoretical basis (jYcm), can scale to larger models (gfVU), provides straightforward baselines for comparison (gfVU), addresses the important issue of toxicity mitigation (7CV4), and shows significant detoxification results across various baselines and benchmarks (zPHB).

During the discussion, the main concerns raised were about adding more baselines (jYcm, gfVU, 7CV4), more benchmarks (jYcm, 7CV4), an ablation study of the hyperparameter (jYcm, 7CV4), and providing examples of detoxifying text (jYcm, gfVU, 7CV4). In response, we have incorporated additional baselines (Table 1 and 2, Figure 2), benchmarks (Table 1 and 2), hyperparameter ablation studies (Table 5), and examples of detoxifying text (Appendix B.3) into our revised paper.
Overall, all updates, including minor changes to the paper caused by additional experiments during the discussion period, are reflected in Tables 1, 2, 4–9, Figures 2 and 3, and Appendices A.2–A.4 and B.1–B.4.
We hope that these new experiments and analyses, along with the clarifications provided in our individual responses, address all the concerns raised.

Once again, we thank all the reviewers for your insightful and invaluable feedback, which has substantially improved our work.

Sincerely,

The Authors

---

### Meta-Review · Area_Chair_hY7c · 2024-12-21

**Metareview:**

This paper presents UniDetox, a universal detoxification method for large language models (LLMs) that eliminates the need for model-specific tuning. It generates detoxified text through dataset distillation with contrastive decoding, producing distilled text from GPT-2 that can detoxify other models, such as LLaMA-2, without requiring additional hyperparameter adjustments. Experiments demonstrate UniDetox's effectiveness in mitigating toxicity and reducing political bias consistently across models.

Pros:
1. The proposed universally applicable detoxification approach is interesting. It addresses the limitation of existing model-specific methods by utilizing dataset distillation.
2. It reduces the need for model-specific hyperparameter tuning, as evidenced by successful application to different models with consistent results with a single hyperparameter configuration.
3. Complete evaluation across a range of commonly used detoxification methods from the references. Significant results based on the PT and EMT metrics across different benchmarks.

Cons:
1. Lacking comparison with alternative alignment algorithms such as DPO or other detoxification baselines using task arithmetic between a base model and a tuned model. There is also a lack in comparison with more recent methods. An efficient method to benchmark against would be the cited LM Steering work.
2. The need for different UNIDETOX configurations impacting performance lacks sufficient justification or analysis.
3. UniDetox demonstrates a drop in diversity metrics compared to other methods. A broader evaluation on other LM tasks would help validate its capability to preserve language quality.

This seems a borderline paper to me. There are several key weaknesses identified by the reviewers, where some of them are addressed during the rebuttal phase. Overall, I feel the reasons to accept may outweigh reasons to reject.

**Additional Comments On Reviewer Discussion:**

This seems a borderline paper to me. There are several key weaknesses identified by the reviewers, where some of them are addressed during the rebuttal phase. Overall, I feel the reasons to accept may outweigh reasons to reject.

---

### Decision · Program_Chairs · 2025-01-22

Accept (Poster)